# Parameter-Free State Estimation Based on Kalman Filter with Attention Learning for GPS Tracking in Autonomous Driving System

**DOI:** 10.3390/s23208650

**Published:** 2023-10-23

**Authors:** Xue-Bo Jin, Wei Chen, Hui-Jun Ma, Jian-Lei Kong, Ting-Li Su, Yu-Ting Bai

**Affiliations:** 1Artificial Intelligence College, Beijing Technology and Business University, Beijing 100048, China; jinxuebo@btbu.edu.cn (X.-B.J.); 2130062047@st.btbu.edu.cn (W.C.); kongjianlei@btbu.edu.cn (J.-L.K.); sutingli@btbu.edu.cn (T.-L.S.); baiyuting@btbu.edu.cn (Y.-T.B.); 2China Light Industry Key Laboratory of Industrial Internet and Big Data, Beijing Technology and Business University, Beijing 100048, China

**Keywords:** trajectory tracking, state estimation, Kalman filter, Transformer, long- and short-term memory network

## Abstract

GPS-based maneuvering target localization and tracking is a crucial aspect of autonomous driving and is widely used in navigation, transportation, autonomous vehicles, and other fields.The classical tracking approach employs a Kalman filter with precise system parameters to estimate the state. However, it is difficult to model their uncertainty because of the complex motion of maneuvering targets and the unknown sensor characteristics. Furthermore, GPS data often involve unknown color noise, making it challenging to obtain accurate system parameters, which can degrade the performance of the classical methods. To address these issues, we present a state estimation method based on the Kalman filter that does not require predefined parameters but instead uses attention learning. We use a transformer encoder with a long short-term memory (LSTM) network to extract dynamic characteristics, and estimate the system model parameters online using the expectation maximization (EM) algorithm, based on the output of the attention learning module. Finally, the Kalman filter computes the dynamic state estimates using the parameters of the learned system, dynamics, and measurement characteristics. Based on GPS simulation data and the Geolife Beijing vehicle GPS trajectory dataset, the experimental results demonstrated that our method outperformed classical and pure model-free network estimation approaches in estimation accuracy, providing an effective solution for practical maneuvering-target tracking applications.

## 1. Introduction

GPS tracking is a crucial field in autonomous driving and machine learning applications, such as motion robotics [1,2], computer vision [3,4], motion training [5,6], autonomous driving [7,8], and so on. GPS tracking uses state-of-the-art estimation methods to estimate the actual trajectory from measurements. The Kalman filter (KF) [9], a popular state estimation method, is a recursive filter designed to estimate the state and signal of a system from noisy, limited, and incomplete measurements. It has several advantages, such as accurately estimating the system’s state and signal in the presence of incomplete measurements and uncertainty. Additionally, the Kalman filter is computationally efficient, easy to implement, and widely used in various systems, such as navigation, communication, control, signal processing, and machine learning.

Although the Kalman filter is effective in many situations, it has some limitations. The Kalman filter assumes that the noise follows a Gaussian distribution and relies on model parameters. In practical autonomous driving applications, especially at levels four and five, there are various challenging environments that can affect GPS data accuracy. These environments include areas with high buildings, dense trees, long tunnels, multilayer junctions, underpasses, and bridges. Additionally, severe weather conditions such as snow, wet roads, old roads, grassy areas, foggy weather, and roads that have been shoveled can further impact the accuracy of GPS data. In these situations, the GPS data often experience different types of noise, including colored noise and outliers. Obtaining accurate statistical characteristics for these noise types can be difficult, and this in turn affects the accuracy of the filters used to process the GPS data.

As for the nonlinear estimation problem, researchers have proposed a series of nonlinear Kalman filters, such as the extended Kalman filter [10] (EKF), the unscented Kalman filter [11] (UKF), etc. The EKF calculates a Jacobi matrix to linearize nonlinear system models using first-order Taylor expansion. Therefore, it is only appropriate for low nonlinearity. The UKF obtains nonlinear equations by performing an unscented transformation on sampling points, approximating the probability density function of nonlinear states. The UKF’s estimation accuracy can be within the second order and it has performed better than the EKF. However, similarly to the EKF, the UKF must transfer the state covariance matrix, where mathematical operations can cause the covariance matrix to lose its positive definiteness and symmetry, leading to filter failure.To address the limitations of the EKF and UKF, a particle filter [12], also known as the Monte Carlo filter, was introduced. The particle filter is a non-parametric approach that uses a set of particles to approximate the posterior distribution of the state variables.

Therefore, researchers have proposed various improved filters such as model-free, particle filter, wavelet transform, Gaussian mixture model, adaptive filtering, etc. The different filters have pros and cons that must be considered and adjusted based on the specific application requirements. Model-free filters and particle filters [13] are two popular improved filters. The former method does not require an a priori model and can adaptively estimate the statistical characteristics of noise, showing better robustness. For example, based on the expectation maximization (EM) algorithm [14], Shumway and Stoffer proposed an algorithm for estimating Kalman filter parameters using the EM algorithm [15]. Another EM algorithm proposed combines the Kalman filter for linear dynamic system state estimation and the EM algorithm for parameter estimation [16]. The latter, through simulation and approximating the posterior probability distribution through sampling, can handle nonlinear and non-Gaussian problems and is more flexible and robust. Although model-free and particle filters can improve the robustness and resilience, they have issues with accuracy and computational efficiency.

In addition, wavelet transform filters, Gaussian mixture model filters, and adaptive filters are also widely used. The wavelet transform filter [17] is primarily suitable for static signal processing and has limitations for signals with a non-stationary behavior. The Gaussian mixture model filter [18] requires the specification of prior and mixture model counts, and its computational complexity is high. The adaptive filter [19] does not require the establishment of an accurate model, but it still requires some prior knowledge or empirical parameters. Therefore, it is necessary to select a filter method based on the specific application needs and to perform a comprehensive analysis and optimization to improve the performance of these algorithms. In summary, despite their unique benefits, these filters have shortcomings regarding their adaptability, computational complexities, and modeling assumptions.

The above filters are based on model parameters. This means they must use prior knowledge to guarantee filtering performance, as they do not fully utilize the measurement data. As previously mentioned, severe environments and weather in autonomous driving will complicate the statistical characteristics of GPS data, and this system model information will be included in the measurement data. Using measurement big data in filtering algorithms is becoming increasingly crucial and practical, and this can capture complex patterns and relationships among hidden variables, providing high-dimensional information for data modeling. Additionally, it can train machine learning models, such as artificial neural networks, capable of learning highly nonlinear and complex relationships. By leveraging big data and modern machine learning techniques, more efficient filtering algorithms can be developed, to handle the complexity and diversity of real-world data streams. Hence, it is crucial to study the use of big data for feature extraction and develop improved filtering methods, to increase accuracy and efficiency.

Leveraging the vast amount of sensor-collected big data can greatly enhance our understanding of synthesized signals and facilitate the identification of environmental noise [20]. This underscores the significance of utilizing such datasets for feature extraction. By harnessing these extensive and intricate datasets, it becomes possible to develop various filters or enhance the effectiveness and performance of the existing ones. Since these datasets are typically large-scale and intricate, leveraging advanced machine learning techniques like neural networks becomes crucial for improving filtering performance. These powerful algorithms enable the analysis and processing of sensor-collected big data, leading to the extraction of more accurate features, such as periodicity, high-frequency components, and mixed noise. The combination of sensor-collected big data and modern machine learning techniques can thus drive the advancement of filtering algorithms, enabling them to effectively handle the diversity and complexity of real-world scenarios.

Researchers have started investigating network-based estimation methods based on the views mentioned above. In recent years, deep learning networks have been widely used in trajectory tracking, utilizing their powerful modeling capabilities and ability to extract potential features from the data. Recurrent neural networks (RNNs) [21] are commonly used for analyzing time-series data, as they are able to capture the dependencies between data points. The long short-term memory (LSTM) network, a variant of the RNN network, was introduced by Hochreiter et al. (1997) [22]. LSTMs employ a complex architecture, incorporating a hidden state unit to determine the importance of input data. This additional layer enhances the network’s ability to capture long-term dependencies by preventing the vanishing gradient problem frequently encountered in regular RNNs. To effectively overcome the issue of gradient explosion, LSTMs employ distinct activation functions and calculation methods, such as the use of gates (e.g., input gates, forget gates, and output gates), which regulate the flow of information through the network, enabling it to model complex, long-term sequences with greater accuracy. LSTMs have become increasingly popular in various fields, such as natural language processing, speech recognition, and image recognition.

Chang-hao Chenet al. [23] proposed IONet, to investigate motion estimation using IMUs based on the Bi-LSTM model, which has a more accurate tracking effect than traditional pedestrian dead reckoning (PDR). The extended nine-axis IONet proposed by Won-Yeol Kim et al. [24] improves trajectory tracking using both gravitational acceleration and geomagnetic data from the IMU. By reducing the data based on the original six-axis IONet input, the estimation accuracy of the position change is improved. Rui-peng Gao et al. [25] proposed a method to track a vehicle in real-time using a temporal convolutional network (TCN), to obtain historical data from the IMU in a mobile phone, when a GPS signal is unavailable.

Deep learning methods have a good learning capability, but in the era of big data, neural networks often need a large amount of input information [26]. This information usually contains a lot of useless or less valuable data, making it difficult for the neural network to learn, reducing the learning efficiency of deep learning methods, and potentially falling into overfitting, which remains a challenge for complex target tracking problems. Recently, the attention mechanism [27] has been used to solve this problem. The attention mechanism can be seen as a simulation of human attention; that is, humans can pay attention to valuable information, while ignoring useless information. A new attention mechanism architecture, a transformer based on the self-attention Seq2seq [28] model, has demonstrated powerful capabilities in sequential data processing, such as natural language processing [29], audio processing [30], and even computer vision [31]. Unlike a RNN, transformer allows the model to access any part of history without considering the distance, making it more suitable for mastering repeated patterns with long-term dependencies and for preventing overfitting. Currently, applying this framework to model tracking has attracted extensive attention from researchers.

Based on the aforementioned disadvantages, traditional estimation methods are usually based on statistical principles, with relatively low computational requirements and a lower requirement for an accurate number of samples. However, conventional estimation methods may not be able to account for nonlinear relationships among multiple variables. On the other hand, deep learning methods can handle large-scale datasets using techniques such as neural networks and can discover more complex patterns and extract critical features. Moreover, deep learning models can adaptively learn relationship patterns in a dataset, reducing the risk of dependence on human rules and prior knowledge. Therefore, combining deep learning with traditional estimation methods can find a balance between model interpretability and high accuracy, thereby improving model performance and robustness, making this more suitable for practical problems.

However, the existing methods rely on predefined parameters and have difficulty with complex nonlinear models. This article intends to use neural networks to learn parameters and address the limitations of traditional methods. To better estimate the trajectory of maneuvering targets, this paper proposes a new tracking method, to solve the problems faced in practice by estimation methods: learning measurement data using a mechanism containing transformer multi-headed self-attention and LSTM, to obtain the statistical properties of its complex motion; at the same time, based on the output of the network, the EM method is used to estimate the dynamics and measurement characteristics of moving targets, and real-time system parameter modeling is carried out in the estimation, to provide more accurate model parameters for the Kalman filter. After learning, the EM algorithm can obtain more real-time and accurate system parameters and improve the trajectory tracking estimation accuracy.

The remainder of this paper is structured as follows: In Section 2, we provide a comprehensive description of the proposed model, which includes a detailed explanation of the network model structure, the mathematical background of the EM algorithm, the Kalman filter, and the overall model framework. Specifically, we present the design of different components, including the input layer, hidden layer, and output layer, as well as the training process for the network. Additionally, we delve into the EM algorithm, which is used to estimate the parameters of the deep learning model, and the Kalman filter, which is applied to smooth the hidden states of a dynamic system. In Section 3, we present the dataset used in the experiment, describe the experimental environment, detail the contents and evaluation metrics of the experiment, and finally analyze the results. Specifically, we describe the characteristics, size, sources of the dataset, and experimental setup, including the software, hardware, and preprocessing steps. In Section 4, we provide the conclusions of the paper and future directions for research.

## 2. Parameter-Free State Estimation

Generally, the linear system model is as follows:(1)$$\begin{array}{cc} \; & x_{k}=Ax_{k-1}+\omega_{k}\\ \; & y_{k}=Cx_{k}+\nu_{k}\\ \; & \omega_{k}∼N\left(0,Q\right),\nu_{k}\left(0,R\right)\\ \; & x_{0}∼N\left(m_{0},P_{0}\right),k=1,2,\ldots ,N \end{array}$$
where $x_{k}$ is the state, $y_{k}$ are the measurement data, *A* is the State transform matrix, *Q* is the state noise, $\omega_{k}$ is the state noise covariance, *C* is the measurement matrix formula, *R* is the measurement noise, $\nu_{k}$ is the measurement noise covariance, $m_{0}$ and $P_{0}$ are the mean and covariance of the initial state, respectively. Usually, the above parameters can be modeled based on historical knowledge or system mechanisms. (The use of linear models can reduce the model complexity, simplify algorithm derivation and calculation, and make problems easier to model and solve, which is beneficial for explaining the principles of the algorithm.)

The Kalman filter is the optimal autoregressive data processing algorithm, and the Kalman filtering process is as follows:(2)$$\begin{array}{cc} \; & {\hat{x}}_{k|k-1}=A_{k}{\hat{x}}_{k-1|k-1}\\ \; & P_{k|k-1}=A_{k-1}P_{k-1|k-1}A_{{}_{k-1}}^{T}+Q_{k-1} \end{array}$$
and,
(3)$$\begin{array}{cc} \; & {\hat{x}}_{k|k}={\hat{x}}_{k|k-1}+K_{k}\left[y_{k}-C_{k}{\hat{x}}_{k|k-1}\right]\\ \; & K_{k}=P_{k|k-1}C_{k}^{T}{\left[C_{k}^{T}P_{k|k-1}C_{k}+R_{k}\right]}^{-1}\\ \; & P_{k|k}=\left[I-K_{k}C_{k}P_{k|k-1}\right] \end{array}$$

In the above equation, *K* is the Kalman filter gain. It is clear from the above equation that the Kalman filtering algorithm consists of two steps. Equation (Equation 2) is the estimation process, where the estimates ${\hat{x}}_{k|k}$ are estimated from the previous one ${\hat{x}}_{k|k-1}$ obtained. Equation (3) is the update process, the estimation is based on the filtering gain of the update.

The classic Kalman filter method relies on accurate model parameters, which can be challenging to obtain in practical applications. Additionally, the simplified linear model used in the Kalman filter may have a significant discrepancy from the actual nonlinear model, due to the inherent nonlinear characteristics of real-world systems. This discrepancy becomes evident in the measurement data collected from the system.

In this paper, we address this issue by incorporating measured data to complement the linear ideal model mentioned in Equation (Equation 1). Specifically, we utilize a deep neural network to train the relationship between the measured data and the reference state offline. This approach allows us to capture the nonlinearities present in the system model and bridge the gap between the simplified linear model and the actual system. The overall system model and the integration of the deep neural network are illustrated in Figure 1.

The proposed model structure in this paper consists of two steps: (1) Offline training: The offline training involves training a system model learner based on LSTM with attention, using the system’s initial parameters (Figure 2). By combining the encoding attention structure of a transformer and LSTM, the model learns from observation data without modeling system dynamics and measurement characteristics, capturing the motion characteristics of the system through neural network training. (2) Online estimation: During estimation, measurement data are input into the attention LSTM learning module to obtain accurate dynamic characteristics, the online part employs the EM algorithm to update the model parameters in real-time. Kalman filtering is then used for online recursive estimation (Figure 3).

### 2.1. Offline Training System Learner

The transformer’s ability to capture long-term dependencies is due to its integration of multi-head self-attention and residual connections, which enhances the model’s training depth and reduces the risk of overfitting. The combination of LSTM and a transformer encoder enhances the structural advantages and sequence modeling ability of the transformer encoder (shown as Figure 2). This model can leverage the known information of the system’s mechanistic model and also obtain the complex dependency relationship of the system from historical measurement data through training the network model.

Multi-headed self-attention [28] generates multiple attentions in the network, acting on features separately and in parallel, with the input to the module being the observed data $y_{1:N}=\left[y_{1},y_{2},\ldots ,y_{N}\right]$, the attention task-related query vector represented as Query, and information such as the input features represented as key-value pairs, Key and Values, respectively. Query, Key, and Values are obtained by linearly varying the input measurements $y_{1:N}$. The process of attentional action can be represented as follows:(4)$$Query=W_{Q}y_{1:N}$$
(5)$$Key=W_{K}y_{1:N}$$
(6)$$Value=W_{V}y_{1:N}$$
(7)$$Attention\left(Query,Key,Value\right)=\mathrm{softmax}\left(\frac{QueryKey^{T}}{\sqrt{d_{k}}}\right)Value$$
where $W_{Q}$, $W_{K}$, and $W_{V}$ are trainable parameter projection matrices and $d_{k}$ is the feature dimension of Key. According to the input Query and Key, the data can be used to obtain the dot product, the weights corresponding to the elements of the input data Value can then be obtained from the Softmax function. $d_{k}$ is used to scale the dot product, to prevent the obtained dot product from being too large, which is conducive to rapid learning.

The essence of the multi-headed self-attendance mechanism is a linear transformation after stitching together the results of multiple attention calculations. This mechanism allows the model to use different feature information obtained at different locations, thus increasing the diversity of features. The multi-headed self-attention is calculated as shown below.
(8)$$Mutil-HeadAttention\left(Query,Key,Value\right)=Concat\left(head_{1},\ldots ,head_{t}\right)W^{o}$$
(9)$$head_{i}=\mathrm{softmax}\left(\frac{Query_{i}Key_{i}^{T}}{\sqrt{d_{k}}}\right)Value_{i}\;i=1.\ldots t\;$$
where *t* is the total number of heads,$W^{o}$ is the weight matrix for ensuring that the target dimension is met, Concat is the splicing operation of the vector, and $head_{i}$ denotes the features of the *i* header.

In this paper, a transformer encoder structure is used to encode and learn the underlying dynamical properties of the observed data, using the multi-headed attention mechanism to construct higher dimensional, multi-channel attentional information and uncover rich information. The attention-based learning module feeds the observed data into the transformer encoder module, which encodes the location of the observed data and feeds the encoded data into the multi-headed self-attentive layer. To prevent network degradation and accelerate convergence, the encoder is also structured with residual connections, layer-norm layers, and feed-forward neural networks. We output potential coding features through the transformer encoder module.
(10)$${\tilde{y}}_{transformerEn\mathrm{coder}}=TransformerEn\mathrm{coder}(y_{1:N})$$

After the transformer, we use the LSTM to learn longer dependencies between the measurement data.
(11)$${\tilde{y}}_{lstm}=LSTM\left({\tilde{y}}_{TransformerEncoder},h_{t-1}\right)$$
where $h_{t-1}$ indicates the hidden state at the previous moment.

The system model learner (as shown in Figure 2) achieves the learning of system dynamics through offline training. The input–output pairs for learning are the observed data and the reference state of the system. The transformer encoder structure with multi-head self-attention and LSTM networks are utilized to learn the latent features of the data. This approach eliminates the need for modeling system dynamics and measurement characteristics. First, the transformer encoder learns the long-term dependencies of the observed data through a self-attention mechanism. Then, the LSTM network further enhances the modeling ability of the sequence data. The combination of the two can effectively extract the features required for parameter estimation. The goal of this deep neural network is to learn the dynamic characteristics of the system, and its output will be input into the EM algorithm as observation data.

### 2.2. Online Kalman Filtering Based on EM Algorithm

The EM algorithm uses the observation data output from deep neural networks to estimate parameters such as the state transition matrix, observation matrix, and noise covariance matrix required for Kalman filter.

After offline training, the network model still deviates from the actual system. During the estimation process, the online estimation part uses the EM algorithm to update the model parameters in real time. Specifically, the measurement data are input into the trained attention LSTM learning module, to obtain accurate dynamic characteristics data that characterize the motion sequence. Then, online recursive estimation is performed through Kalman filtering. Furthermore, the EM algorithm is used to update the current state-transition matrix, measurement matrix, state noise, measurement noise variance, and other parameters (as shown in Figure 3).

The left side of Figure 3 shows the EM algorithm, according to Jenson’s inequality, we have
(12)$$\begin{array}{cc} logp\left(y_{1:N}|\theta \right) & =\log(\int q\left(z_{0:N}\right)\frac{p(z_{0:N},y_{1:N}|\theta )}{q(z_{0:N})}dz_{0:N})\\ \; & \geq \int q\left(z_{0:N}\right)\log\frac{p(z_{0:N},y_{1:N}|\theta )}{q(z_{0:N})}dz_{0:N} \end{array}$$
where $z_{0:N}$ is the hidden variable, θ is the unknown parameter to be estimated, and $\theta^{\left(n\right)}$ denotes the parameter at the nth iteration. The PDFs of the latent variables are
(13)$$q\left(z_{0:N}\right):=p\left(z_{0:N}|y_{1:N},\theta^{\left(n\right)}\right)$$

Substituting (13) into (12), we obtain
(14)$$\begin{array}{cc} \; & \int p\left(z_{0:N}|y_{1:N},\theta^{\left(n\right)}\right)\log\frac{p(z_{0:N},y_{1:N}|\theta )}{p(z_{0:N}|y_{1:N},\theta^{\left(n\right)})}dz_{0:N}\\ \; & =\int p\left(z_{0:N}|y_{1:N},\theta^{\left(n\right)}\right)\log p\left(z_{0:N},y_{1:N}|\theta \right)dz_{0:N}\\ \; & -\int p\left(z_{0:N}|y_{1:N},\theta^{\left(n\right)}\right)\log p\left(z_{0:N}|y_{1:N},\theta^{\left(n\right)}\right)dz_{0:N} \end{array}$$

It can be seen that the latter term of the above equation is not relevant to θ and can be omitted. For state estimation, maximization of the measurement $\log p(z_{0:N},y_{1:N}|\theta )$ of the PDF is equivalent to maximization of the log likelihood of Q.
(15)$$Q\left(\theta ,\theta^{\left(n\right)}\right)=E\left[\log p\left(z_{0:N},y_{1:N}|\theta \right)|y_{1:N},\theta^{\left(n\right)}\right]$$

The EM algorithm starts with an initial estimate $\theta^{\left(0\right)}$ and then iterates over the E and M steps of Algorithm 1.
**Algorithm 1** EM algorithm**Input**:
Initial estimation of parameters $\theta^{\left(0\right)}$, error ε, maximum iteration number $n_{m}$**Output**:
$\theta^{\left(*\right)}$1:**repeat**2:E-step: compute $Q(\theta ,\theta^{\left(n\right)})$3:M-step: $\theta^{\left(n+1\right)}\leftarrow \arg\max_{\theta }Q\left(\theta ,\theta^{\left(n\right)}\right)$4:**until**$\theta^{\left(n+1\right)}-\theta^{\left(n\right)}|<\varepsilon$, or iteration number is up to $n_{m}$**return**$\theta^{*}\leftarrow \theta^{\left(n+1\right)}$

Let the state to be estimated for the system be $x_{k}$ and the measured value be $y_{k}$, with the probability distribution of
(16)$$\begin{array}{cc} \; & p\left(x_{k}|x_{k-1}\right)=N\left(x_{k}|Ax_{k-1},Q\right)\\ \; & P\left(y_{k}|x_{k}\right)=N\left(y_{k}|Cx_{k-1},R\right)\\ \; & p\left(x_{0}\right)=N\left(m_{0},P_{0}\right) \end{array}$$

Therefore, by extending the probability distribution function in Equation (14) to an exponential function, we obtain
(17)$$p\left(x_{k}|x_{k-1}\right)=\exp\left\{-\frac{1}{2}{\left[x_{k}-Ax_{k-1}\right]}^{T}Q^{-1}\left[x_{k}-Ax_{k-1}\right]\right\}{\left(2\pi \right)}^{-u/2}|Q{|}^{-1/2}$$
(18)$$p\left(y_{k}|x_{k}\right)=\exp\left\{-\frac{1}{2}{\left[y_{k}-Cx_{k}\right]}^{T}R^{-1}\left[y_{k}-Cx_{k}\right]\right\}{\left(2\pi \right)}^{-v/2}|R{|}^{-1/2}$$
(19)$$p\left(x_{0}\right)=\exp\left\{-\frac{1}{2}{\left[x_{0}-m_{0}\right]}^{T}{P_{0}}^{-1}\left[x_{0}-m_{0}\right]\right\}{\left(2\pi \right)}^{-u/2}|P_{0}{|}^{-1/2}$$
where *u* and *v* are the dimensions of the state transfer matrix and measurement matrix, respectively.

Under the assumptions of the Markov property of states and the conditional independence of measurements, the current state is only related to the previous state, and the current observation is only related to the current state.Based on the above assumptions, the joint probability distribution can be obtained:(20)$$p\left(x_{0:N}|y_{1:N}\right)=p\left(x_{0}\right)\prod_{k=1}^{N}p(x_{k}|x_{k-1})\prod_{k=1}^{N}p(y_{k}|x_{k})$$

Therefore,
(21)$$\begin{array}{cc} \ln p(x_{0:N}|y_{1:N}) & =-\sum_{k=1}^{N}\left(\frac{1}{2}{\left[y_{k}-Cx_{k}\right]}^{T}R^{-1}\left[y_{k}-Cx_{k}\right]\right)-\frac{N}{2}\ln\left|2\pi R\right|\\ \; & \;-\sum_{k=1}^{N}\left(\frac{1}{2}{\left[x_{k}-Ax_{k-1}\right]}^{T}Q^{-1}\left[x_{k}-Ax_{k-1}\right]\right)-\frac{N}{2}\ln\left|2\pi Q\right|\\ \; & \;-\frac{1}{2}{\left[x_{0}-m_{0}\right]}^{T}{P_{0}}^{-1}\left[x_{0}-m_{0}\right]-\frac{1}{2}\ln\left|2\pi P_{0}\right| \end{array}$$

Combining (21) and (15), substitute the probability density function of the hidden variable $z_{0:N}$ into Jensen’s inequality, and note that the parameter independent term in the density function can be omitted. We can obtain the expected form of the parameter:(22)$$\begin{array}{cc} Q(\theta ,\theta^{\left(n\right)}) & =-\frac{1}{2}\ln|2\pi P_{0}|-\frac{N}{2}\ln|2\pi Q|-\frac{N}{2}\ln\left|2\pi R\right|\\ \; & -\frac{1}{2}tr\left\{{P_{0}}^{-1}\left[P_{0|N}+\left(m_{0|N}-m_{0}\right){\left(m_{0|N}-m_{0}\right)}^{T}\right]\right\}\\ \; & -\frac{1}{2}\sum_{k=1}^{N}tr\{Q^{-1}\left[\left(x_{k}-Ax_{k-1}\right){\left(x_{k}-Ax_{k-1}\right)}^{T}|y_{1:N}\right]\}\\ \; & -\frac{1}{2}\sum_{k=1}^{N}tr\{R^{-1}\left[\left(y_{k}-Cx_{k}\right){\left(y_{k}-Cx_{k}\right)}^{T}|y_{1:N}\right]\} \end{array}$$

Based on the maximum value theory, calculate the partial derivative of (22) and obtain the parameters:(23)$$\frac{\partial Q(\theta ,\theta^{\left(n\right)})}{\partial \theta^{\left(n\right)}}=0$$

Utilizing the offline training system learner mentioned in Section 2.2, we can capture the long-term dependence between data and the potential correlations between data, which can better estimate the parameters of the filter model. Based on the linear least squares method, the observations and state estimators output by the neural network are constructed into a linear regression model, and the state transition matrix *A* and observation matrix *C* are obtained by solving the covariance matrix of the estimators, then there are
(24)$$A=\left(\sum_{k=1}^{N-1}\left[{\hat{x}}_{k|k}{\hat{x}}_{k-1|k-1}^{T}\right]\right){\left(\sum_{k=1}^{N-1}\left[{\hat{x}}_{k-1|k-1}{\hat{x}}_{k-1|k-1}^{T}\right]\right)}^{-1}$$
(25)$$C=\left(\sum_{k=0}^{N-1}y_{lstm,k}{\left[{\hat{x}}_{k|k}\right]}^{T}\right){\left(\sum_{k=0}^{N-1}\left[{\hat{x}}_{k|k}{\hat{x}}_{k|k}^{T}\right]\right)}^{-1}$$
where ${\hat{x}}_{k|k}$ denotes the estimate of the filter at step *k*.

Then, based on A,C, statistical modeling based on the error of state estimators and calculating the covariance matrix of state noise *Q*, construct a statistical model of observation noise using the covariance matrix of residuals and obtain the covariance matrix of observation noise *R*:(26)$$\begin{array}{cc} Q & =\frac{1}{N-1}\sum_{k=0}^{N-2}(\left[{\hat{x}}_{k+1|k+1}\right]-A\left[{\hat{x}}_{k|k}\right]){\left(\left[{\hat{x}}_{k+1|k+1}\right]-A\left[{\hat{x}}_{k|k}\right]\right)}^{T}\\ \; & +AVar\left[{\hat{x}}_{k|k}\right]A^{T}+Var\left[{\hat{x}}_{k+1|k+1}\right]\\ \; & -Cov\left({\hat{x}}_{k+1|k+1},{\hat{x}}_{k|k}\right)A^{T}-ACov({\hat{x}}_{k+1|k+1},{\hat{x}}_{k|k} \end{array}$$
(27)$$R=\frac{1}{N}\sum_{k=0}^{N-1}\left[y_{lstm,k}-C\left[{\hat{x}}_{k|k}\right]\right]{\left[z_{k}-C\left[{\hat{x}}_{k|k}\right]\right]}^{T}+CVar\left[{\hat{x}}_{k|k}\right]C^{T}$$

Using recursive output states ${\hat{x}}_{k|k}$ through Kalman filtering, the estimated trajectory is obtained. When using Kalman filter, *A* is the state-transition matrix calculated using (24), *Q* is the state noise covariance calculated using (26), *C* is the measurement matrix formula (25), and *R* is the measurement noise covariance calculated using Formula (27).

## 3. Experiment

### 3.1. Experiment Set-Up

To construct the network models, we utilized the Pytorch deep learning framework as an open-source library. All experiments described in this paper were conducted on a personal computer with an Intel Core i5-7300HQ @2.50 GHz CPU processor and 8 GB RAM. In our experiment, we used an LSTM model with 64 hidden vectors and a self attention network with 8 heads as encoders, while training the model through the Adam algorithm.The neural network model was initialized with default parameters in Pytorch, which included weight initialization for deterministic deep learning networks. Furthermore, the random number seed was fixed on the computer for all experiments, to ensure result reproducibility.

To comprehensively evaluate the forecasting performance of the different models, this study employed four evaluation metrics. These metrics were the root mean square error (RMSE), mean absolute error (MAE), symmetric mean absolute percentage error (SMAPE), mean square error (MSE), and Pearson correlation coefficient (P). The formulas for these four evaluation metrics are shown in Equations (28)–(32).
(28)$$\mathrm{RMSE}=\sqrt{\frac{1}{n}\sum_{k=1}^{n}{\left({\hat{x}}_{k|k}-x_{k|k}\right)}^{2}}$$
(29)$$\mathrm{MAE}=\frac{1}{n}\sum_{k=1}^{n}\left|({\hat{x}}_{k|k}-x_{k|k}\right)|$$
(30)$$\mathrm{SMAPE}=\frac{100\%}{n}\sum_{k=1}^{n}\frac{|\left({\hat{x}}_{k|k}-x_{k|k}\right)|}{\left(|{\hat{x}}_{k|k}|+|x_{k|k}|\right)/2}$$
(31)$$\mathrm{MSE}=\frac{1}{n}\sum_{k=1}^{n}{\left({\hat{x}}_{k|k}-x_{k|k}\right)}^{2}$$
(32)$$P=\frac{\sum_{i=1}^{n}\left(x_{k|k}-{\bar{x}}_{k|k}\right)\left({\hat{x}}_{k|k}-{\bar{\hat{x}}}_{k|k}\right)}{\sqrt{\sum_{i=1}^{n}{\left(x_{k|k}-{\bar{x}}_{k|k}\right)}^{2}\sum_{i=1}^{n}{\left({\hat{x}}_{k|k}-{\bar{\hat{x}}}_{k|k}\right)}^{2}}}$$
where *n* is the total number of samples in the dataset, $x_{k,k}$ is the *k*-th true value of the reference trajectory, ${\bar{x}}_{k,k}$ is the mean of the true values of the reference trajectory, ${\hat{x}}_{k,k}$ is the *k*-th estimate of the estimated trajectory obtained by the model, and ${\bar{\hat{x}}}_{k|k}$ is the average of the estimated trajectories obtained by the model. The first three of the four evaluation indicators mean that the smaller the resulting value, the more accurate the model estimation. The significance of the evaluation indicator R is that the larger the result value, the better the fit of the reference trajectory to the estimated trajectory.

### 3.2. Results and Discussion

(1) Case 1: In this case, the trajectory path dataset utilized for simulation was obtained from the work of btbuIntelliSense (https://github.com/btbuIntelliSense, accessed on 20 October 2023). A total of 4200 trajectory paths were provided, each consisting of 201 data points. Each data point included the horizontal and vertical coordinate values of longitude and latitude, which were used to simulate the real movement path. The simulated data spanned a two-dimensional plane ranging from 0 to 100, with the addition of pink noise signals. The noise was generated utilizing a MATLAB signal processing toolbox function known as “ColoredNoise”. In order to evaluate the improvement in filter performance using parameter self-learning modules, the selected were compared with various traditional Kalman filter models, including the Kalman filter along with the process models CA [32], CV [33], Singer [34], current statistical [35], and EM Kalman filter [16]. At the same time, we also compared the combination methods of neural networks and filtering models, including the experimental results of transformer-EF and LSTM-EF, which will be of great benefit to our research, as shown in Table 1.

We conducted a comprehensive comparison of the trajectory tracking performance of the different filter models, and the results are presented in Table 2. The proposed model exhibited the lowest error evaluation metric values compared to the other models, reflected by the highest *p*-value, RMSE, MAE, SMAPE, MSE, and *p* values of 1.71, 1.35, 4.36, 2.92, and 0.99, respectively.

Compared with the other five filter models, our proposed model had the advantage of not requiring complex manual parameter adjustment processes. This adjustment process typically requires extensive domain knowledge and is difficult to optimize. Compared with the other two neural network combined filter models, our proposed model effectively improved the estimation accuracy. In comparison to the other seven models, our model achieved a reduction in RMSE estimation accuracy of 20.5%, 34.5%, 30.4%, 70.2%, 72.1%, 29.3%, and 10.9%, respectively. The MAE was also reduced by 10.0%, 11.7%, 10.6%, 59.8%, 61.9%, 19.1%, and 3.6%, respectively. Furthermore, the MSE was reduced by 36.7%, 57.1%, 51.7%, 91.9%, 92.2%, 50.1%, and 20.8%, respectively. Additionally, our model resulted in an increase in SMAPE by 27.1%, 33.6%, 30.6%, 68.3%, 57.5%, 6.6%, and 3.3%, respectively.The proposed model’s largest *p* value from the index signifies an excellent estimation accuracy, indicating the strongest correspondence between the estimated and true values.

Our proposed model was found to have the best fit between the estimated values and true values, as indicated by the P-index. The experimental data analysis confirmed that our model outperformed the other filter models in terms of the estimation accuracy and result fit. Specifically, we evaluated the effectiveness of integrating neural networks with filtering models and found that this combination enhanced the performance of our proposed model by refining the estimated parameters of the EM algorithm. This successful combination of neural networks with filter algorithms demonstrated the applicability of our model in the field of trajectory tracking, compared to the traditional methods (Figure 4 and Figure 5).

(2) Case 2: For this case, we utilized the Geolife Beijing vehicle GPS trajectory dataset (https://www.gpsvisualizer.com/, accessed on 20 October 2023), which consists of 13,987 GPS coordinates, where the GPS coordinates are composed of longitude and latitude values. Among these, 80% were chosen as the training set for the model, while the remaining 20% were used as the test set. Figure 6 showcases an example trajectory from this dataset.

Considering the complexity of actual scenarios, we chose some of the latest literature methods for end-to-end deep learning, to comprehensively compare the advantages of our method.

We applied the proposed model to the Beijing vehicle trajectory dataset. Given the intricacy of real-life situations, we meticulously selected a few state-of-the-art end-to-end deep learning modules. Our goal was to holistically evaluate the merits of our approach through comparative experiments, which involved five other models, namely LSTM [22], GRU [36], CNN-LSTM [37], ConvLSTM [38], and PFVAE [39]. For each of these comparison models, we set specific parameters. As an example, for the GRU model, we used 50 hidden neural units, a learning rate of 0.0001, 1000 iterations, and a batch size of 30. The GRU model consisted of two network layers, each containing 50 hidden units. In terms of the sliding window strategy, we applied the same recursive estimation approach as the filter model. The predicted results of all six models utilized in the experiment are presented in Table 2.

**Table 2 sensors-23-08650-t002:** Evaluation of the different filter estimation results for Beijing vehicle GPS motion track data.

Model	RMSE	MAE	SMAPE	MSE	P
LSTM [22]	3.84	3.97	3.75	14.75	0.99
GRU [36]	6.83	5.52	4.29	46.64	0.99
CNN-LSTM [37]	5.92	4.47	3.87	35.04	0.99
ConvLSTM [36]	5.76	4.69	3.15	33.17	0.99
PFVAE [39]	2.52	1.97	1.90	6.35	0.99
The proposed method	1.82	1.32	0.97	3.31	0.99

As shown in Table 2, our proposed TL-EF model showed significant performance improvements compared to the LSTM, GRU, CNN-LSTM, ConvLSTM, and PFVAE models. Specifically, the TL-EF model achieved 52.6%, 73.8%, 69.2%, 68.4%, and 27.7% reductions in RMSE, and 66.7%, 76.1%, 70.5%, 71.8%, and 32.9% reductions in MAE, respectively, when compared to these models. Furthermore, the MSE was reduced by 77.6%, 92.9%, 90.6%, 90.0%, and 47.8%, respectively. The improvement rates for SMAPE were 74.1%, 77.3%, 74.9%, 69.2%, and 48.9%, respectively. Additionally, the TL-EF model obtained P indicators that were comparable to those of the LSTM, GRU, CNN-LSTM, ConvLSTM, and PFVAE models.

In Figure 6, a scatter plot of the estimated and true values for each model trajectory is presented. The estimated values align closely with the true longitude and latitude values of the corresponding scattered points, demonstrating the accurate estimation results of the model. In Figure 7, we provide scatter plots of distance errors for the different estimation models in GPS Euclidean. Euclidean distance refers to the linear distance between two points in 2D, and the Euclidean distance error represents the degree of proximity between the estimated position and the actual position, thus reflecting the accuracy and accuracy of the positioning estimation system. Our proposed model had less fluctuation in the error scatter plot, which verified that our proposed model had a better estimation accuracy. To ensure objectivity, we independently repeated each model 20 times using different random seeds on the Beijing vehicle GPS motion trajectory dataset and recorded the RMSE value. In Figure 8, we used a violin chart to display the statistical results, and the results showed that our proposed method had a smaller error fluctuation range, a more uniform error distribution, and higher estimation accuracy and stability compared to the other models. Based on the above analysis, it can be concluded that the proposed model exhibited a higher accuracy and stability in estimating GPS trajectories, surpassing the traditional and deep learning models. This makes it a valuable application in the field of trajectory tracking. The model holds great significance for state estimation methods and provides reliable technical support for further research.

## 4. Conclusions and Future Work

GPS tracking is a critical research field with numerous applications, including autonomous driving, unmanned vehicles, and drones. However, it faces several challenges, particularly when dealing with highly maneuverable targets or complex measurement data containing colored noise. For instance, the GPS navigation signal received by users’ terminals may be too strong, due to high dynamic Doppler shift, or weak, due to occlusion, multipath, flicker, and other factors. Hence, there is an urgent need to develop novel and effective estimation algorithms to overcome these challenges and enhance the accuracy of state estimation.

To address this challenge, this paper proposed a Kalman estimation method that incorporates a learning module based on transformer and LSTM. By combining a transformer encoder structure with LSTM, a neural network parameter learning module was designed to capture the motion characteristics of the system through learning offline state measurement data, eliminating the need to model system dynamics and measurement characteristics. Each model’s parameters for estimating Kalman filter estimates are then obtained using the EM algorithm based on the neural network parameter learning module’s output. This approach enables accurate state estimation for parametric position systems using Kalman filters.

In the future, further research could explore the utilization of different deep learning networks or fine-tune the trade-off between estimation accuracy and computational complexity, to enhance the practicality of state estimation systems. By addressing these challenges, we could unlock new possibilities for advancing and deploying state estimation systems in various applications.

## Figures and Tables

**Figure 1 sensors-23-08650-f001:**
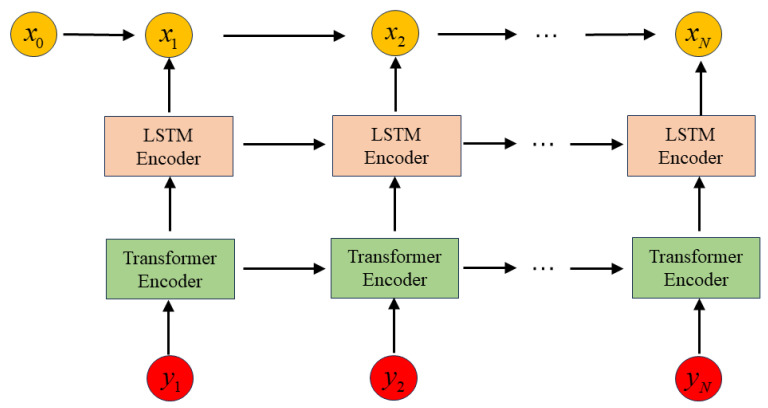
System Model Learner.

**Figure 2 sensors-23-08650-f002:**
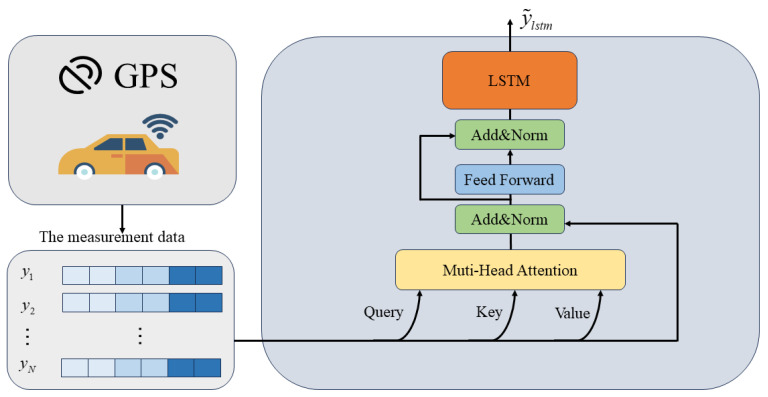
Offline training system learner based on attention and LSTM.

**Figure 3 sensors-23-08650-f003:**
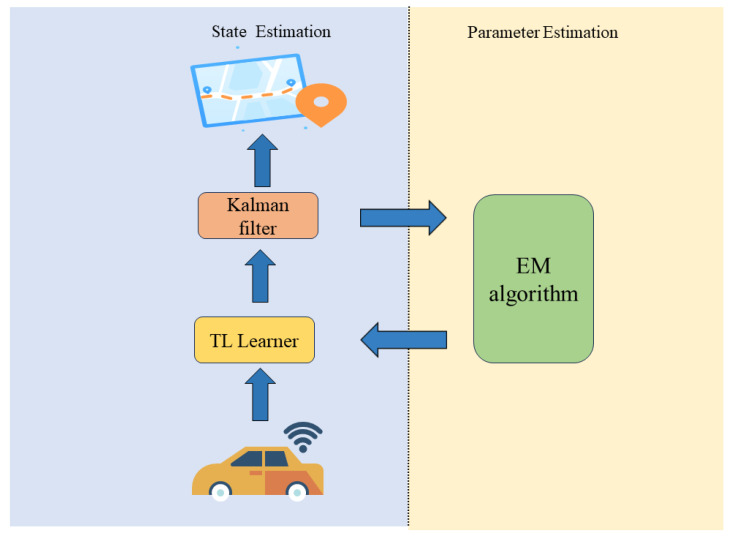
Online Kalman filtering based on EM.

**Figure 4 sensors-23-08650-f004:**
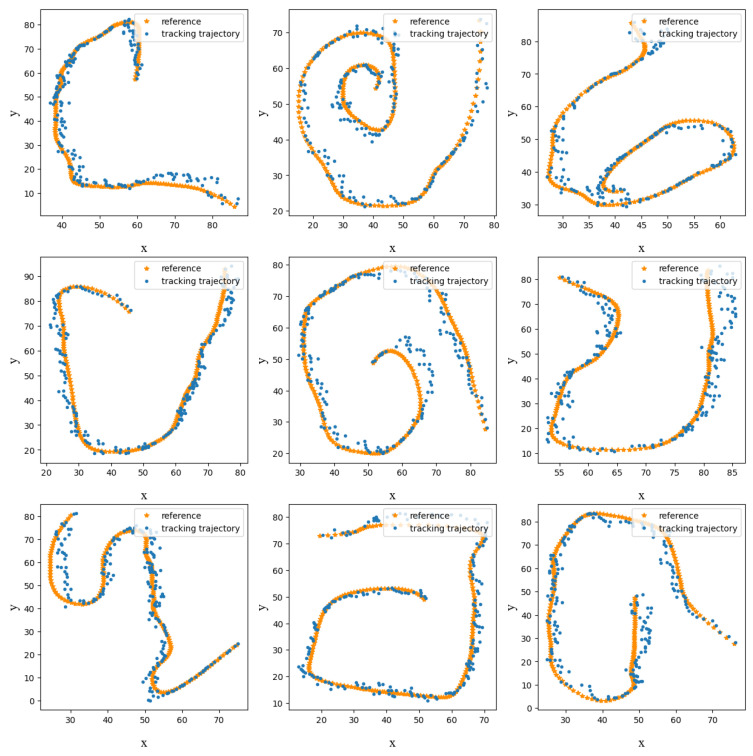
Examples of tracking results.

**Figure 5 sensors-23-08650-f005:**
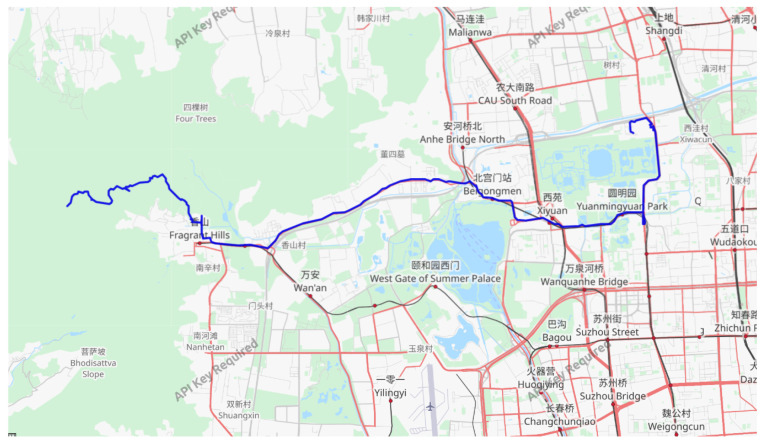
An example of the Geolife Beijing vehicle GPS motion track dataset.

**Figure 6 sensors-23-08650-f006:**
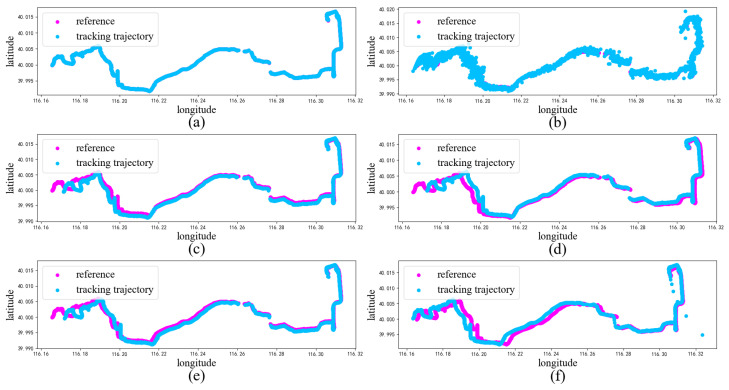
Comparison of the tracking results for GPS trajectories: (**a**) trajectory estimated using the proposed method, (**b**) trajectory estimated using PFVAE, (**c**) trajectory estimated using LSTM, (**d**) trajectory estimated using GRU, (**e**) trajectory estimated using CNN-LSTM, (**f**) trajectory estimated using ConvLSTM.

**Figure 7 sensors-23-08650-f007:**
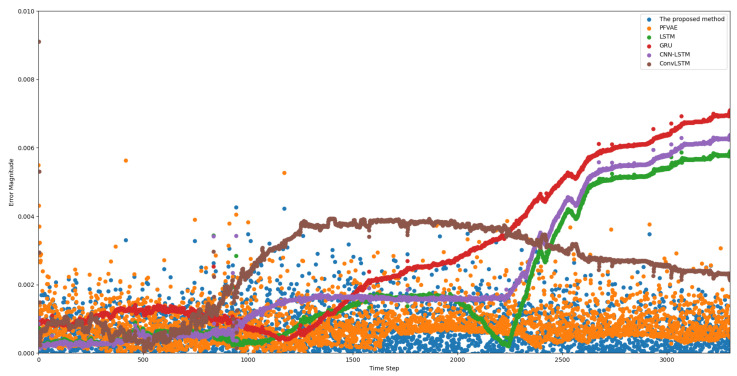
Scatter plot of Euclidean distance errors in GPS estimations using the different estimation models.

**Figure 8 sensors-23-08650-f008:**
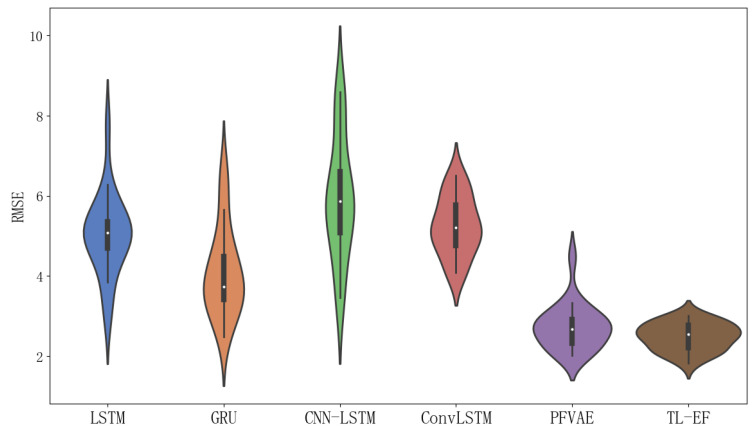
RMSE violin plot for each model with 20 independent repetitions of the experiment on the Beijing vehicle GPS motion trajectory dataset.

**Table 1 sensors-23-08650-t001:** Evaluation of the different filter estimation results for the simulation trajectory dataset.

Model and Its Parameters		RMSE	MAE	SMAPE	MSE	P
Kalman filtering with CA [32]	T = 0.5 Mv = 400 Qq = 100	2.83	1.64	7.13	8.00	0.98
	T = 1 Mv = 1600 Qq = 200	2.15	1.50	5.98	4.62	0.99
Kalman filtering with CV [33]	T = 0.5 Mv = 400 Qq = 100	3.18	1.62	7.00	10.11	0.97
	T = 1 Mv = 1600 Qq = 200	2.61	1.53	6.57	6.81	0.98
Kalman filtering with Singer [34]	T = 0.5 Mv = 400 Qq = 100	3.24	1.64	7.10	10.49	0.98
	T = 1 Mv = 1600 Qq = 200	2.46	1.51	6.29	6.05	0.98
Kalman filtering with Current Statistical [35]	T = 0.2 Mv = 50 xamax = 50	5.76	4.30	15.00	33.18	0.93
	T = 0.1 Mv = 25 xamax = 30	5.74	3.36	13.76	32.95	0.93
Kalman filtering with EM [16]		6.14	3.55	10.28	37.69	0.98
Transformer-EF		2.42	1.67	4.67	5.85	0.99
LSTM-EF		1.92	1.40	4.51	3.69	0.99
The proposed method		1.71	1.35	4.36	2.92	0.99

Parameters: The parameter T in the table indicates the time step used to calculate the state-transition matrix A and the process covariance matrix Q. The parameter Qq adjusts the process noise covariance matrix Q to modify the level of uncertainty associated with the process noise. The variable Mv represents the variance of the measurement noise. The parameter xamax is used to adjust the state-transition model to reflect specific system characteristics or constraints, in order to fine-tune the response speed and noise sensitivity of the filter to meet different application requirements.

## Data Availability

The data in the experiment used to support the findings of this study are available from the corresponding authors upon request.

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
