# Peer review of "Parameter-Free State Estimation Based on Kalman Filter with Attention Learning for GPS Tracking in Autonomous Driving System"

_sensors, 2023, doi:10.3390/s23208650_

Round 1
Reviewer 1 Report
This article proposes a Kalman filtering algorithm that combines the transformer mechanism and LSTM to enhance GPS tracking accuracy. The algorithm learns the system's motion characteristics through offline network training, enabling precise estimation of the positional state. The article integrates the traditional Kalman filter with more accurate temporal prediction models from neural networks, yielding precise outcomes in experiments. However, the article still has some issues:
1. Sections 2.1 and 2.2 have identical titles.
2. While the method of utilizing transformer and LSTM for parameter network learning yields reasonable parameters from simulation or real data, the model itself lacks distinctiveness. There might be room for original contributions rather than relying solely on general networks for parameter tuning and training.
3. The four evaluation metrics presented in the article may not comprehensively compare the tracking performance among models. Introducing additional evaluation metrics could potentially offer a more comprehensive assessment.
4. While the article presents tracking accuracy comparisons between the proposed method and traditional filtering techniques, it would be beneficial to include results from experiments that combine neural networks and filtering models more extensively. This would provide a more comprehensive demonstration of the proposed method's tracking accuracy.
Major revision
Reviewer 2 Report
It is necessary to expand practical research in various climatic conditions, since this issue is very relevant.
Reviewer 3 Report
Summary:
The paper proposes a new automatic parameter estimation mechanisms for Kalman
Filters. The approach is based on a combination.
Evaluation:
The paper provides an interesting idea. However, neither the explanation of
the idea nor the experimental section is fully convincing. The idea of
combining a LSTM and a Transformer (or at least parts of it) seems redundant
as Transformers are a generic replacement for LSTMs. Unfortunately, the
authors did not provide an evaluation of a Transform-only approach for
comparison. The next issue is the lack of explanation of the system model used
in the evaluation part. This coincides with the ambiguity regarding the
description of the Kalman Filter. All descriptions indicate a linear Kalman
Filter, however the typical model used for cars involve a non-linear system
model. This is never explained or cleared in the paper. Finally, the
evaluation lacks a statistical analysis, as every experiment is only executed
once, however, multiple runs with different seeds are necessary to evaluate the
performance of highly randomized training algorithms as used to train any kind
of neural network.
Consequently, I suggest a major rework of the paper to extend the necessary
explanations as well as the justifications for the approach. Additionally,
additional experiments should be conducted to clearly indicate the benefit of
a combined LSTM and Transformer model against a pure transformer model. All
evaluations should also be extended with a statistical analysis of multiple
runs with different seed to show stability and robustness of the approaches.
Detailed Comments:
- Kalman Filter vs Extended and Unscented Kalman Filter
- Line 31: Precise measurements <-> Noise
- Line 46: Redundant EM-KF indicates already EM + KF
- Line 62: Adaptive Filter is Model-Free, Model-Free filter should be model-free ...
- Line 99: LSTMs are in the process of beeing replaced by transformers because of better training performance
- Eq 1: Linear Equation for non-Linear Problem? Mix up of Noise and Covariance in v_k
- Sec 2.1: The relation between the output of the Neural Network and the EM Algorithm and the Kalman Filter is not explained well
- Sec 2.2: Expected structural advantages of the combination of LSTM and Transformer need to be explicitly stated!
- Equation 13: the last Q^(n) should be subscript.
- Equations 15 to 17 needs an explanation or source, why they can be deduced
from 14
- The same goes for the inference of 18 from 15 to 17
- Equation 19: ln should be not italic to indicate the logarithm of p. The R
and Q terms miss the exponents u/2 and v/2 of equations 15 and 16, The last
part is missing a terminating vertical line.
- Equation 20: The derivation of this equation from 19 and 13 is non-trivial
and should be explained
- Line 253: the measurement does not need to be square, therefore a single
variable u is not enough to describe the shape.
- Equations 22,23,24 and 25 need explanation of their deduction
- Equation 28: seems incorrect as squares are used, which are not used in any
of the original formulations
- Section 3:
- Used System Model of Kalman Filter and References
- Description of data and data structure contained in the datasets
- Definition of R Value is missing
- Comparison with a pure transformer model is missing
- (Hyper-)parameter deduction is missing, parameters are stated, but not deduced, explained or justified
- Explanation of the different References in Case 1 and Case 2 is necessary.
- Line 32 - 37: Overly complex sentence
- Line 41-43: No Explanation of Particle Filters ⇾ Explanation in Line 47
- Line 43: robustness and robustness
- Line 75 - 85: too vague and repetitive
- Line 167 - 175 does not understandably explain the relation between the linear system model and the non-linear system
Behavior.
Round 2
Reviewer 3 Report
Detailed Comments:
5: The performance of the evaluated NN-Architectures: LSTM, Transformer and LSTM+Transformer is not convincing without providing detailed information on the internal structure of each network. LSTM+Transformer may win the competition because of larger model size compared to the more simple ones.
6: Missed the point. The considered problem typically is a non-linear one. Approaching a non-linear problem with a linear approach needs an explicit linearization step with appropriate analysis of the linearization error. EKF and UKF are non-optimal filters for non-linear problems; however, they are generally better suited than linear Kalman-Filters.
Please provide a justification, why your problem is linear or include a comparison of a linear KF and EKF or UKF on your problem.
8: The statement is too vague. Is there any explainable structural advantage of adding the LSTM compared to increasing the size/complexity of the Transformer?
21: The Hyperparameters related to the model parameters of the used NN-Models as well as the used training algorithms. For example, the amount of NN-Layers and their feature-Sizes as well as the used training algorithm (SGD, Adam, AdamW) with the training parameters.
22: The evaluation misses any statistical analysis regarding the stochastic nature of the used NNs. Typically, NN training needs to be conducted multiple times to evaluate the performance.
Line 48-50: Repeated Statement
The language improved, but I suggest an additional proof-reading run.
